Geographic context affects the landscape change and fragmentation caused by wind energy facilities

Diffendorfer Jay E. jediffendorfer@usgs.gov 1
Dorning Monica A. 1
Keen Jolene R. 1 2
Kramer Louisa A. 1
Taylor Robert V. 1
1 Geosciences and Environmental Change Science Center, United States Geological Survey , Lakewood , CO , United States of America
2 American Association of Geographers , Washington , DC , United States of America
Jones Roger
Electronic publication date: 2019 Jul 16
Publication date: 2019
Volume: 7
Electronic Location ID: e7129
Received 2019 Feb 19; Accepted 2019 May 15
Copyright year: 2019
License: This is an open access article, free of all copyright, made available under the Creative Commons Public Domain Dedication. This work may be freely reproduced, distributed, transmitted, modified, built upon, or otherwise used by anyone for any lawful purpose.
License URL: https://creativecommons.org/publicdomain/zero/1.0/

Keywords: Core area, Habitat loss, Connectivity, Roads, Wind energy, Fragmentation, Habitat fragmentation, Geographic context

Funding: Land Change Science program The Energy Mission Area at the United States Geological Survey Funding for the project came from the Land Change Science program and the Energy Mission Area at the United States Geological Survey. The funders had no role in study design, data collection and analysis, decision to publish, or preparation of the manuscript.

==============================
Wind energy generation affects landscapes as new roads, pads, and transmission lines are constructed. Limiting the landscape change from these facilities likely minimizes impacts to biodiversity and sensitive wildlife species. We examined the effects of wind energy facilities’ geographic context on changes in landscape patterns using three metrics: portion of undeveloped land, core area index, and connectance index. We digitized 39 wind facilities and the surrounding land cover and measured landscape pattern before and after facility construction using the amount, core area, and connectivity of undeveloped land within one km around newly constructed turbines and roads. New facilities decreased the amount of undeveloped land by 1.8% while changes in metrics of landscape pattern ranged from 50 to 140%. Statistical models indicated pre-construction development was a key factor explaining the impact of new wind facilities on landscape metrics, with pre-construction road networks, turbine spacing, and topography having smaller influences. As the proportion of developed land around facilities increased, a higher proportion of the facility utilized pre-construction developed land and a lower density of new roads were built, resulting in smaller impacts to undeveloped landscapes. Building of new road networks was also a predictor of landscape fragmentation. Utilizing existing development and carefully placing turbines may provide opportunities to minimize the impacts of new wind energy facilities.

Introduction

Generating electricity from wind is a leading technology for meeting energy demand while minimizing greenhouse gas emissions and air pollution. The global installed capacity of wind energy reached 539 GW by the end of 2017 (World Wind Energy Association, 2018) and forecasts suggest continued growth and market penetration doubling to quadrupling levels of installed onshore wind energy in the next 20 to 30 years (Hand et al., 2012; US Energy Information Administration, 2015; International Energy Agency, 2016). For example, the International Energy Agency (2016) suggests the installed capacity of wind energy will grow from 6% of the 6393 GW (or 384 GW) of total global energy capacity in 2014 to 13% of 11,170 GW (1526 GW) in 2040. While increasing wind energy generation may reduce emissions, it will also result in landscape change associated with developing new facilities, having potentially negative effects on wildlife and other ecosystem functions.

Wind facility footprints can affect wildlife species through changes in the amount, quality, and configuration of habitat, especially through the addition of new roads. The amount of habitat remaining in a landscape can have strong effects on overall species richness and persistence (Rosenzweig, 1995), and the size and location of habitat patches can affect species abundances, behavior, and persistence through edge effects, and other ecological processes (Diffendorfer, Gaines & Holt, 1999; Gibson et al., 2013). A rich and large literature exists documenting the impacts of roads on ecosystems and their functions, creating a subfield of research called “road ecology” (Forman et al., 2002). Roads affect species directly via habitat loss, roadkill, and behavioral avoidance, as well as indirectly through modifications to the abiotic environment and conveyance of chemicals and noise (Forman & Alexander, 1998; Coffin, 2007). Wind turbines transform relatively small percentages of the land areas they occupy; however, the roads between turbines add to the total land transformation associated with a facility and cause extensive changes in landscape configuration, fragmenting remaining wildlife habitat. Thus, the road networks associated with facilities are a fundamental mechanism causing changes to terrestrial ecosystems.

A key question is how can new wind facilities be sited and designed to maximize energy production while minimizing potential negative impacts to the environment associated with the facility footprint? A number of countries plan new wind facilities with the goal of minimizing environmental consequences. For example, the European Union has guidelines associated with Natura 2000 that describe a ‘strategic planning’ approach to wind energy development (European Commission, 2013), and the United States developed voluntary siting guidelines (Wildlife Service, 2016) which recommended a tiered evaluation of locations for new facilities. Both of these approaches include considerations of habitat loss and fragmentation caused by the wind facility. Understanding what drives the levels of land transformation, fragmentation, and road network expansion associated with new facilities could inform siting decisions.

Both the roads and the total amount of land transformed by facilities have been estimated and compared across different types of energy (Fthenakis & Kim, 2009; Evans & Kiesecker, 2014). Estimates of the land transformation from wind facilities vary by an order of magnitude (Diffendorfer & Compton, 2014) but we do not know why. We currently have not fully investigated the causes of variation in levels of land transformation (measured as ha/MW of installed capacity) and road network expansion across facilities. The few studies that have investigated this variation suggest the amount of transformation depends on factors intrinsic to the turbines themselves and on the geographic context in which a facility is sited (Denholm et al., 2009; Jones & Pejchar, 2013; Diffendorfer & Compton, 2014). Jones & Pejchar (2013) showed individual wind turbines had fewer impacts at locations with more pre-construction development at the scale of one km around each turbine. Diffendorfer & Compton (2014), showed geographic context influenced the total amount of land transformation caused by wind facilities, with facilities on tilled or flat landscapes requiring less land than those in forested locations or on hills and mesas. These studies focused primarily on the drivers of the total land area developed by a facility and did not assess drivers of landscape fragmentation or expansion of road networks. Determining why the levels of land transformation, fragmentation, and roads vary across facilities may allow the siting of wind energy that minimizes levels of landscape change, and perhaps impacts on terrestrial species.

We investigated how new wind facilities affect the loss and fragmentation of undeveloped lands with a focus on the new roads built during construction. In particular, we ask if factors associated with the turbines themselves (size and spacing), topography, and pre-construction land use affect the levels of impact caused by facilities. Based on previous results by Jones & Pejchar (2013) and Diffendorfer & Compton (2014) we expected facilities placed in landscapes dominated by human activity to utilize existing road networks, resulting in fewer new roads and a higher proportion of new infrastructure on developed land. Thus, sites with more pre-construction development would have lower amounts of loss and fragmentation of undeveloped lands. We also expected larger turbines would be spaced farther apart, and these greater distances between turbines would require a larger road network. Similarly, facilities with more turbines would require more roads. These factors may or may not affect fragmentation metrics as greater turbine spacing or more turbines would also increase the total extent of a facility, a potential confounding factor for some metrics. Finally, we expected facilities at locations with more topographic relief to require larger road networks, as roads follow topography. We expected these sites to have higher levels of loss and higher fragmentation of undeveloped lands than sites with less topographic relief.

Materials & Methods

We selected 39 facilities across the United States, sampling across gradients in topography, turbine capacity, land cover, and turbine string configuration (for facility selection details and a table describing the facilities see Diffendorfer & Compton (2014)). We did not develop a fully orthogonal study design because some combinations of the stratifying variables did not exist. For example, there were no three MW facilities in flat and forested locations. We digitized development associated with each facility as well as pre-construction land cover. We then used an information-theoretic modeling approach to understand how various facility, topographic, and pre-construction characteristics influenced change in landscape fragmentation at each site.

Digitizing procedures

The digitizing process included a number of steps. First, we digitized the surface development, road networks, and turbine locations created during the installation of facilities (Diffendorfer & Compton, 2014). Second, we delineated the study extent at each facility by buffering the newly constructed hardened surface road network and turbine locations by one km and then merged and dissolved these into a single layer. Third, we manually digitized the pre-construction roads and land cover within the study extent for each facility.

We considered the buffer distance for determining the study extent and how it could affect the measured impacts of new facilities. As buffer size increases, so does the area of the study extent, making the relative area impacted by the wind facility smaller. For example, imagine studying a single turbine with 2ha of development surrounded by undeveloped land. The proportion of the study site occupied by the development changes as the buffer around it increases (0.23 at 200 m, 0.038 and 500 m, and 0.009 at one km). We selected a one km buffer for three reasons. First, to study the role of geographic context on development, we wanted a spatial scale that was large enough to adequately represent the pre-construction land cover. Second, buffers larger than one km began to create oversized study extents, with wind facilities making up a diminishingly small amount of the land surface. Third, Jones & Pejchar (2013) used a one km buffer, allowing us to make comparisons between studies using the same spatial scale.

We used high-resolution digital photographs to map both pre-construction and post-construction roads using on-screen digitizing methods. We started with existing road data (from either national, state, or county sources) and then digitized additional roads based on one m resolution USDA/NRCS Digital Orthophoto Quad Imagery (DOQ) mosaics in ArcMap 10.4 and 10.5 (Esri, Redlands, CA, USA). We selected imagery with the year nearest to the initial operational date of the wind facility for both pre and post construction roads. For example, to digitize pre-construction roads with a facility that began operations in June 2010 and 2007, 2009 and 2011 imagery was available, we used the 2009 imagery immediately prior to June 2010. Similarly, the imagery closest to, yet after, June 2010 was used to digitize post-construction roads. In this case, if 2010, 2012, and 2014 imagery were available, we used 2012 imagery. When DOQs were not available, we used the ESRI “imagery” basemap (which is not dated) available in ArcMap and matched this to Google Earth data imagery to attain a year.

In some instances, we observed other roads in the digital imagery that were not mapped in existing road datasets. We included these when (1) they were wider than three m and (2) vegetation had not grown back in the areas (tracks) that would be driven over by tires. We assumed roads less than three m wide with vegetation regrowth were either not used or used infrequently and would have relatively small impacts compared to larger roads with more traffic. We mapped all pre-construction roads widened during construction but considered these part of the pre-construction conditions.

We required a simplified, consistent land-cover classification across all facilities, despite the broad geographic and ecological range found within the 39 facilities. We reclassified the 20 land-use types in the 2011 National Land Cover Dataset (NLCD) (Homer et al., 2015) into ‘undeveloped’ and ‘developed’, and retained the original open water classification. Undeveloped lands included NLCD categories of deciduous, evergreen, and mixed forest, shrub/scrub, grassland/herbaceous, woody, and emergent herbaceous wetlands. Developed lands included NLCD categories of cultivated crops, pasture/hay, developed open space, developed low, medium, and high intensity, and barren land. The remaining NLCD classifications did not overlap with the facility extents. In the NLCD, grassland/herbaceous areas include grasslands supporting grazing, while pasture/hay areas are specifically planted and managed to support livestock and the production of hay crops. In our classification, undeveloped lands represent “natural vegetation”, which is potential habitat for species that would utilize a region’s natural, undeveloped vegetation types. At Texas wind farms this might represent Blackland prairies or woodlands, while in Pennsylvania, this might represent deciduous forest. Because habitat is species-specific, we did not attempt to link the observed changes in land cover and roads to impacts on species. Given this approach, we considered pasture lands and developed open space as “developed” because they do not represent undeveloped vegetation types. Though these land-cover types might support some native species, we assumed they would not likely support species who require undeveloped vegetation types.

The reclassified 30 m resolution of NLCD data did not accurately match the imagery of developed, undeveloped, and water land-cover types within the one km facility extents. To improve our estimation of pre and post construction land cover, we changed the raster to a two m resolution and manually edited the reclassified NLCD data to more closely conform to high-resolution DOQs using the ARIS grid editor (ARIS, 2017).

Finally, the digitized pre and post construction roads were merged into the land-cover maps by adding roads into the ‘developed’ classification. The overall approach resulted in pre and post construction two m resolution raster maps of each facility classified as undeveloped, developed, and water within one km extents as well as maps of the pre and post construction road and turbine infrastructure. These GIS files and the data used in the analyses are available at the USGS ScienceBase data catalog (https://doi.org/10.5066/F7639NZB).

Analysis

We used linear models and beta regressions to estimate how the geographic context at a facility, such as topography and levels of pre-construction development, turbine characteristics, and the newly installed roads influenced three landscape metrics measuring aspects of habitat loss and fragmentation (described below). We also used the same approach to estimate how geographic context and turbine characteristics influence new road construction. We initially performed graphical and exploratory analysis of the data, and present paired t-tests describing changes in the number of patches, road length, and the total area of undeveloped land. We also examined correlations between our explanatory variables and used these in model development.

Landscape metrics related to loss and fragmentation

The (n = 39) study sites varied in extent and the amounts of undeveloped land, thus we selected three landscape metrics shown to be independent of the abundance of a land cover type, yet sensitive to levels of aggregation (Wang, Blanchet & Koper, 2014). These three metrics were the dependent variables in our statistical models. The first metric, the change before versus after construction, in the proportion of undeveloped land (area of undeveloped land/total extent) measured the amount of potential habitat affected by the facility. Second, the change in the core area index (CAI) of undeveloped land, assesses the core area of remaining patches as a percentage of the total area of undeveloped land using an area-weighted mean (McGarigal, Cushman & Ene, 2012). CAI required an edge depth for determining how far into a patch edge effects attenuate. Because species vary in their edge responses, we analyzed 50 m, 100 m, and 200 m edge depths, found similar results, and report results for 50 m edge depth. For the third metric, we modeled change in the connectance index (CI) for undeveloped land. CI is the proportion of the total connections between patches that occur within a threshold set distance (McGarigal, Cushman & Ene, 2012). For example, in a 4-patch system, six possible connections exist between the patches (1–2, 1–3, 1–4, 2–3, 2–4, and 3–4). If three of these were within the specified distance then CI = 3/6 = 0.50. Similar to CAI, we analyzed the change in CI at threshold distances of 100, 200, and 300 m and found similar results across all three. We report the 100 m results. Though not included in our models due to dependence on area (Wang, Blanchet & Koper, 2014), we also describe changes in the number of patches pre versus post construction. All landscape metrics were calculated using FRAGSTATS on the pre and post-construction raster layers. We initialized FRAGSTATS to consider undeveloped land as the classification of interest. As such, patches were defined as groups of undeveloped pixels separated from other groups by developed land.

The selected metrics measured aspects of land cover associated with ecological patterns and dynamics. For example, habitat loss is recognized as a fundamental driver of declines in biodiversity and the abundance of species (Brooks et al., 2002; Cushman, 2006). The size and presence of core areas affect abundance and demography of species requiring a patch size of a minimum area (Burke & Nol, 1998; Laurance et al., 1998; Helzer & Jelinski, 1999). Finally, connectivity affects species movement patterns in fragmented landscapes, their spatial ecology, and persistence (Crooks & Sanjayan, 2006).

Explanatory variables and candidate models

We used an information-theoretic approach to compare models of the effects of geographic context, turbine characteristics, and newly installed roads on the selected landscape metrics (Burnham & Anderson, 2002). All models included the coefficient of variation (CV) of elevation, mean slope, turbine capacity, turbine count, mean nearest neighbor distance between turbines (in m), proportion of disturbed land pre-installation (area of developed land before construction/area total extent), density of roads pre-installation (length of pre-construction roads/area total extent, m/ha), and the density of new roads (length of new roads/area total extent, m/ha). Thus, the models included variables describing topography, turbine characteristics, levels of pre-existing development and roads, and new roads added during construction. The coefficient of variation (CV) in elevation (m) and the mean slope (degrees) were calculated across all cells within the facility extent.

To avoid multicollinearity, we excluded models with mean nearest neighbor distance and either the number of turbines or the mean turbine size from the candidate model set, while retaining all other combinations of these variables. The mean nearest neighbor distance between turbines (m) was correlated (see results) with both the number of turbines at a site and the mean size of turbines (nameplate capacity in MW).

The statistical models included all additive combinations of these variables across the 39 study sites, excluding models with both nearest neighbor distance and turbine size, or nearest neighbor distance and the turbine count. Because our sample size of 39 was small relative to the number of explanatory variables, we limited the candidate model set to models with no more than three explanatory variables but ran models using all possible combinations of three explanatory variables.

We also investigated if the density of new roads added during construction was influenced by the explanatory variables related to topography, turbine characteristics, and levels of pre-installation development. These analyses followed the same methods as those described above.

Model implementation

We fit candidate models in R (R Development Core Team, 2013) using either linear models (LMs) or beta regression (using the betareg package when variables ranged from 0 to 1), and performed model selection with MuMin package (R Development Core Team, 2013). Dependent variables analyzed with beta regression included the proportion of undeveloped land and the proportional change in connectance index. These dependent variables were transformed using y×n−1+0.5n where n is the sample size (39) and y is the dependent variable (Smithson & Verkuilen, 2006). We ranked models using the sample size adjusted AIC (AICc) (Burnham & Anderson, 2002) and model-averaged using AICc weights, which are the log likelihood of the model divided by the sum of the log likelihood across all models. AICc weights are interpreted as the conditional probability that a model is the best model in the candidate model set. We used R2 as an estimate of overall model fit. For the top two ranking models within each candidate model set, we estimated variance inflation factors (VIF’s) for each explanatory variable and found none suggesting severe multicollinearity issues in any of the models (highest VIF across all models was 1.91).

To understand what predictor variables were the most important across the candidate model set we followed recent recommendations and used model-averaged standardized coefficients and their standard errors (Cade, 2015; Galipaud, Gillingham & Dechaume-Moncharmont, 2017) and the relative variable importance (Giam & Olden, 2016). For the model-averaged standardized coefficients, we model-averaged using all models in the candidate model set, substituting zero in those models when the parameter of interest was absent (Burnham & Anderson, 2002).

To estimate model coefficients and effect sizes we used the unstandardized model-averaged estimates, calculated from ‘natural’ or conditional model averaging, where only those models containing the variable were used. We did this because full model-averaged estimates are used to understand the importance of one variable compared to another but can bias estimates of effect sizes downward (Symonds & Moussalli, 2011; Grueber et al., 2011). We used the unstandardized model-averaged estimates of model coefficients, their standard errors, and bi-plots of the dependent variables versus the explanatory variable of interest while setting the other variables in the model to their mean values, to understand patterns in the response and to visually check the modeled relationship. For beta regressions, we estimated confidence bands around the fitted lines by bootstrapping the model selection and model averaging process 5,000 times and taking the 2.5% and 97.5% quantiles.

Results

Overall patterns

The construction of wind facilities creates relatively narrow, linear development driven by the roads and buried cables between turbines, as well as the access road and electric grid connections to the facility (Fig. S1). However, the road network and overall patterns of construction vary markedly across facilities (Fig. S1). In general, these patterns fall within the ‘incision’ and ‘dissection’ phases of fragmentation described by Jaeger (2000), where a linear feature cuts into or completely bisects a landscape.

Across the 39 sites, turbines with higher capacities were spaced farther apart (turbine capacity vs. mean nearest neighbor distance, r = 0.47, p = 0.002). However, sites with more turbines spaced them closer together (turbine count vs. mean nearest neighbor distance, r =  − 0.35, p = 0.031), yet the turbine count was not correlated with turbine capacity (turbine count vs. turbine capacity, r =  − 0.17, p = 0.29) or average blade length (turbine count vs. mean average blade length, r =  − 0.08, p = 0.622).

Before construction, the mean length of roads was 44,731 ± 5,199 m (mean ± SE). The mean length of newly constructed roads was 20,427 ± 2679 m or a mean percent increase averaged across facilities of 52.64 ± 5.9%. The new roads increased the mean number of patches of undeveloped land from 61.31 ± 9.35 to 97.18 ± 10.71 (paired t-test, t = 4.633, df = 38, p < 0.001). However, because some facilities had large increases in the number of patches (e.g., from 28 to 164), the mean percent increase in the number of patches across facilities was 143%. Patch density within the facility extents also increased from 1.75 ± 0.26 to 2.75 ± 0.29 patches/ha (paired t-test, t = 5.194, df = 38, p < 0.001). While roads, number of patches, and patch density increased from ∼50 to ∼140%, the change in the total area of undeveloped land affected by construction was ∼2%. Pre-construction, sites had an average of 2,355.07 ± 369.47 ha of undeveloped land, which was reduced to 2294.17 ± 359.54 ha after construction (paired-test, t = 5.5847, df = 38, p < 0.001), with a mean percent decrease across facilities of −1.89%. This new development (60.92 ± 10.91ha, average for each facility) was ∼7% of the average area of developed land on the sites pre-construction (872.04 ± 182.17 ha).

What drives the levels of land cover change?

Change in undeveloped lands

We modeled the change in the proportion of undeveloped land before and after construction. Model selection uncertainty was high with 15 models within 5 AICc units (Table S1). Pre-construction development, turbine size, the density of new roads, and pre-construction road density had high relative variable support (Table 1), but the model-averaged standardized parameter values were very low for turbine size, density of new roads, and pre-construction road density. The standardized model-averaged estimates and relative variable importance (Table 1) indicated pre-construction development had far greater support than the other variables. Facilities with higher proportions of development prior to construction had smaller changes in the proportion of undeveloped land after installation (Fig. 1).

Table 1 Variable importance and parameter estimates for the statistical models.

Relative Variable Importance using AICc (RVI), model-averaged standardized parameter estimates (MA standarized estimate), and their standard errors (MA standarized SE), the model-averaged parameter estimate (MA estimate) and their standard errors (MA SE) for explanatory variables used in each of the statistical models. Coefficient of variation in elevation (CV elevation), Average slope across facility extent (Slope), Nameplate capacity of turbines (Turbine size), nearest-neighbor distance (NN distance), density of new roads (New roads), proportion of developed land pre-construction (Pre-development), road density pre- construction (Pre-roads).

Model and variables	RVI	MA standardized estimate	MA standardized SE	MA estimate	MA SE	
Change in the proportion of undeveloped land						
Pre-development	1	−0.4615	0.05154	−1.2299	0.1374	
Turbine Size	0.5239	0.0435	0.0493	0.113	0.0486	
New Roads	0.4629	0.038	0.0483	0.0344	0.0158	
Pre-roads	0.3851	−0.0267	0.0404	−0.0081	0.0042	
Slope	0.1464	0.0075	0.0229	0.0122	0.0088	
CV elevation	0.1063	−0.0041	0.0178	−0.0033	0.0034	
No. of Turbines	0.0680	0.0008	0.0105	0.0003	0.0011	
NN Distance	0.0521	0.0017	0.0136	0.0003	0.0004	
Change CAI at 50 m						
Pre-development	1	−2.1373	0.2380	−5.6963	0.6344	
New Roads	0.9988	1.0814	0.2354	0.4534	0.0973	
Pre-roads	0.7155	−0.3693	0.2870	−0.0602	0.0232	
No. of Turbines	0.0833	0.0329	0.1296	0.0115	0.0071	
Slope	0.0709	−0.0261	0.1141	−0.0881	0.0573	
Turbine size	0.0233	−0.0033	0.0395	−0.1916	0.2944	
CV elevation	0.0222	−0.0028	0.0378	−0.0108	0.0186	
NN Distance	0.0182	0.0005	0.0351	0.0002	0.0023	
Change in Connectance Index						
Pre-development	0.9999	−0.9794	0.2064	−2.6107	0.5495	
New Roads	0.8925	0.4798	0.2271	0.2251	0.0685	
Slope	0.6524	−0.2908	0.2433	−0.1067	0.0352	
NN Distance	0.2004	0.0827	0.1821	0.0037	0.0015	
No. of Turbines	0.09	0.0275	0.0996	0.0089	0.0047	
Pre-roads	0.0557	−0.0113	0.0568	−0.0236	0.0163	
Turbine Size	0.0351	0.007	0.0464	0.2676	0.2061	
CV elevation	0.0185	0.0004	0.0246	0.0002	0.0152	
New Road Density						
Pre-development	0.7021	−0.580	0.5048	−2.2028	1.0644	
No. of Turbines	0.5819	0.5658	0.5692	0.0283	0.0117	
CV elevation	0.3576	0.1823	0.3335	0.0431	0.032	
NN Distance	0.3134	−0.2861	0.4808	−0.0081	0.0036	
Slope	0.1542	0.0091	0.1758	0.0141	0.1063	
Pre-roads	0.15	−0.019	0.1528	−0.0148	0.044	
Turbine Size	0.1130	0.0291	0.1462	0.3471	0.4878	

Figure 1 Relationship between the proportion of developed land before the construction of a wind facility and the change in the proportion of undeveloped land before versus after construction.

Points show each facility; the line shows the fitted line from a model-averaged beta regression holding other predictor variables at their mean value. Gray shaded area represents the 2.5% and 97.5% quantiles from 5,000 boot-strapped replicates.

Core areas

The amount of change in Core Area Index (with a 50 m edge depth) after construction was primarily driven by pre-construction levels of development, pre-construction roads, and new roads (Table S1). Standardized model-averaged coefficients and the RVI supported the importance of the three variables and suggested pre-construction development had a larger effect size followed by new road density, and then pre-installation road density (Table 1). Unstandardized model averaged coefficients were large relative to their standard errors for these variables except for pre-construction road density, but even in this case, the 95% confidence interval did not overlap 0 (−0.0602 ± 1.96∗0.0232 =  − 0.106 to −0.015). Increases in pre-construction development (Fig. 2A), and pre-construction road density (Fig. 2B) resulted in smaller changes in CAI. Change in CAI increased as the density of new roads increased (Fig. 2C).

Connectivity

The pre versus post-construction change in the connectance index with a 100 m threshold distance was primarily affected by levels of pre-construction development, new roads, and average slope. In all cases, support for the variables (RVI and standardized model averaged estimates) was relatively high. Furthermore, unstandardized model averaged estimates were large relative to their standard errors so that confidence intervals did not overlap zero for all three variables. The proportional change in CI approached zero as levels of pre-construction development increased (Fig. 3A). As the density of new roads increased, the proportional change in CI increased but with considerable variation (Fig. 3B). For example, at a density of new roads of 7.5 m/ha, the change in the CI ranged from 0 to ∼0.6. Finally, sites with higher average slopes had smaller levels of change in CI, though this relationship had relatively high levels of uncertainty (Fig. 3C). As with the other variables analyzed, pre-construction development had the largest relative effect size on the change in the connectance index, followed by new road density and then average slope (Table 1).

Figure 2 Relationship between the change in the core area index of undeveloped land before versus after construction and three explanatory variables.

Points show each facility; the line shows the fitted linear relationship and 95% confidence interval (gray shaded area) from the model-averaged-model holding other predictor variables at their mean value. (A) The proportion of developed land before the construction of a wind facility, (B) the density of roads before construction, (C) the density of new roads.

Figure 3 Relationship between the proportional change in the connectance index of undeveloped land before versus after construction and three explanatory variables.

Points show each facility; the line shows the fitted line from a model-averaged beta regression holding other predictor variables at their mean value. Gray shaded area represents the 2.5% and 97.5% quantiles from 5,000 boot-strapped replicates. (A) The proportion of developed land before construction. (B) The density of new roads. (C) Average slope.

The role of the proportion of pre-construction development

To better understand the interaction between the development caused by the facility and the levels of pre-construction development, we calculated the proportion of the wind facilities development that intersected pre-construction developed land. The proportion of wind facility developed land in pre-construction developed areas increased nonlinearly as the proportion of pre-construction development increased (Fig. 4, proportion of facility in pre-construction developed lands = 1.93*proportion of developed land pre-construction—0.88* proportion of developed land pre-construction2, R2adj = 0.95, AICc vs linear model = − 71.41 vs. −56.34).

Figure 4 Relationship between the proportion of developed land before the construction of a wind facility and the proportion of the facility built on land that was developed prior to construction.

Points show each facility; the line shows the fitted relationship.

New roads

The density of new roads installed at facilities ranged from 2.03 to 11.8 m/ha. Pre-construction development, the number of turbines, the coefficient of variation in elevation, and the distance between turbines had the highest relative variable importance. However, model-averaged standard errors suggested less support for the coefficient of variation in elevation (Table 1). The density of new roads declined as the proportion of pre-construction development increased (Fig. 5A), increased as the number of turbines at a facility increased (Fig. 5B), declined as the distance between turbines increased (Fig. 5C), and increased as the coefficient of variation in elevation increased (Fig. 5D).

Figure 5 Relationship between the density of new roads after construction and the proportion of developed land before the construction of a wind facility and four explanatory variables.

Points show each facility; the line shows the fitted linear relationship and confidence interval (gray shaded area) from the full model holding other predictor variables at their mean value. (A) The proportion of developed land before the construction of a wind facility. (B) The number of turbines. (C) The mean nearest neighbor distance between turbines. (D) The coefficient of variation in elevation.

Discussion

Our study had four main findings. First, the amount of pre-construction development plays a key role in determining the overall impacts of a wind facility on undeveloped lands. Second, the amount of undeveloped land that is developed during construction is much lower than the level of habitat fragmentation created. Third, new roads play a key role in the levels of change in landscape metrics associated with habitat fragmentation. Fourth, topography and turbine variables had fewer direct effects on the habitat fragmentation metrics, though they did affect new road density.

The amount of pre-construction development was a key predictor of change in all of the landscape metrics and the addition of new roads. Unlike other explanatory variables, the pre-construction density of developed land was included in a predominance of the best-supported models and had high values based on measures of relative variable importance. Scatter plots also indicated relatively strong relationships between pre-construction development and landscape metrics. This result, while not surprising, is still important to confirm. Across the sites, as pre-construction development increased, the amount of undeveloped land declined, and higher proportions of the wind facility utilized developed instead of undeveloped land (Fig. 4), thus reducing the influence of the facility on the loss and fragmentation metrics.

Similar to pre-construction development, the density of newly constructed roads explained variation in the levels of change in the various landscape metrics. While new roads and pre-construction development were correlated, low variance inflation factors in the models for all landscape metrics suggested new roads explained unique variation in the dependent variables. As density of new roads increased at facilities, core areas and connectivity declined, indicating pre-construction patches of undeveloped land were either diminished in size and/or bisected by the new roads.

Across the 39 sites, the proportional change in the area of undeveloped land was much smaller than the proportional change in the number of patches, the CI, and the CAI. Constructing new facilities consumes space, yet the development is essentially long and narrow, spreading out across a network that contains a mix of pre-existing and new roads. The relationship between the pre-existing patterns of undeveloped land and the road influenced-pattern of development caused by the facility creates a process by which area loss is small relative to changes in metrics measuring landscape patterns. This suggests studies that only measuring the levels of land transformation caused by wind facilities, and perhaps other forms of energy, may miss potential impacts caused by road effects and habitat fragmentation.

We note our analyses did not include transmission lines and so only represents facility-level, “on-site” impacts. Department of Energy scenario analyses suggests a 20% expansion of the existing electricity transmission network in the US will be needed to accommodate 404 GW of installed wind energy capacity by 2050 (Department of Energy, 2015). Transmission lines bisect areas and generate disturbance during construction and maintenance and new lines can be controversial (Lienert, Suetterlin & Siegrist, 2015). The effects of transmission lines likely depend on geographic context and vary by species. Above ground transmission lines are a major source of mortality for birds (Loss, Will & Marra, 2014) yet the area under these lines can also support native bees (Wagner, Ascher & Bricker, 2014), nesting birds (Chasko & Gates, 1982; King et al., 2009) and foraging mammals (Takatsuki, 1992).

Siting new facilities

Is it possible to construct facilities that minimize the levels of habitat loss and fragmentation they create? Our results suggest siting facilities in locations with higher levels of pre-construction development and utilizing existing roads may reduce the impacts of the facilities on undeveloped lands. Jones & Pejchar (2013) also found impacts to a variety of indicators declined as both oil/gas and wind energy was sited in locations with more pre-existing disturbance. Facility-level analyses like ours and Jones & Pejchar (2013) add to an empirical basis for larger scale geospatial analyses that examine how much energy can be placed on already disturbed lands (Kiesecker et al., 2011; Fargione et al., 2012; Baruch-Mordo et al., 2019). However, placing energy in already developed areas may be difficult depending on how compatible the energy type is with the land use. Higher levels of pre-construction development occurred because sites were dominated by agricultural land use or rural to semi-rural homes and their associated land cover such as lawns, cleared land, and roads. Wind energy development is often compatible with agriculture and allows energy generation and crop production to coexist on the same landscape. However, not all electricity markets occur near regions dominated by agriculture.

Placing wind energy facilities in semi-rural areas that include a mix of homes and small farms may be more problematic than in agricultural landscapes. These semi-rural areas have higher population densities and thus more individuals may be opposed to wind energy and building a facility may require permission and lease agreements with multiple landowners. In addition, if counties or municipalities regulate road setback distances (to prevent falling blades or towers from blocking roads, hitting homes, etc.), higher road and house densities reduce the number of locations where turbines can be placed (Rogers, Slegers & Costello, 2012).

Our results also indicate other variables, in addition to pre-construction development, might be considered when siting facilities. Lower densities of newly installed roads meant fewer fragmentation impacts. In addition, facilities with more turbines had higher densities of new roads and the density of new roads declined as the distance between turbines increased. This suggests that for a fixed level of overall generation capacity, using fewer but larger turbines, spaced farther apart, may reduce the density of new roads. Roads and buried power lines are necessary at facilities, and turbines are placed to optimize or maximize energy generation. It may be possible to maximize the use of existing roads, and develop optimization approaches to minimize new roads at facilities or the degree of fragmentation they create (Schweitzer et al., 1997; Chung, Bae & Kim, 2016). Best management practices may also be utilized to minimize road impacts, such as collection basins to reduce sedimentation, and reduced travel speeds to minimize road impacts on wildlife.

Clearly, pre-construction disturbance and new roads can influence the effects of new wind facilities on landscape change and fragmentation. However, topography may have a relatively smaller, but perhaps important effect. Our previous analysis (Diffendorfer & Compton, 2014) indicated topography influenced the levels of land transformation, with flatter areas having less transformation than sites with hills. Our new analysis used continuous, not categorical, predictor variables and found relatively weak relationships between landscape metrics and both slope and the coefficient of variation in elevation. Both of these relationships could be explored in more detail as our sample had relatively few facilities in areas with steep, or highly variable, topography.

In practice, our results and those from similar studies could be included in GIS-based siting analyses of wind facilities. A number of studies have considered siting wind or other energy types using GIS, multi-criteria decision methods, or custom software to integrate variables related to energy potential, costs of development, and consequences to humans and natural systems (Tegou, Polatidis & Haralambopoulos, 2010; Siyal et al., 2015; Latinopoulos & Kechagia, 2015; Watson & Hudson, 2015; Milt, Gagnolet & Armsworth, 2016; Sánchez-Lozano, García-Cascales & Lamata, 2016; Rafiee et al., 2018). To be most effective, a geospatial planning approach incorporating road impacts would require the dynamic calculation of roads and their impacts via scenario analyses and perhaps optimization. Similar approaches have been used to optimize the location of best management practices within watersheds (Wang, 2001), and prioritize the removal of hydrological barriers (dams, road crossings) in streams across the great lake region (Moody et al., 2017).

Habitat loss, fragmentation, and species impacts

At the scale of our analysis, new facilities decreased the amount of undeveloped land by an average of ∼1.8% with a maximum value of 4.7%. Though not settled, some studies suggest habitat loss is the main cause of declines in biodiversity and that habitat fragmentation (independent of habitat area) plays a much smaller role (Yaacobi, Ziv & Rosenzweig, 2007; Fahrig, 2013) but see (Hanski, 2015). More research is needed to understand if the habitat loss and fragmentation from wind facilities has negative consequences on species. However, the relatively small amounts of loss of undeveloped land (at one km scales), suggests wind energy may be less likely to generate declines in species richness associated with habitat loss, and more likely to impact a set of species uniquely sensitive to the types of development created by the facility.

Following the hypothesis, and studies supporting it, that urbanization homogenizes ecological communities (McKinney, 2006; Morelli et al., 2016), facilities located in more developed landscapes may have lessened negative ecological effects not only because they utilize less undeveloped land, but also because the pre-construction levels of development have already extirpated species sensitive to habitat loss and habitat fragmentation. Essentially, this hypothesis suggests pre-construction development has already displaced species the new facility may have displaced. In addition, species capable of existing in developed landscapes may not be negatively impacted by wind energy development (a prediction that has not been tested).

Wind facilities, given the need to place turbines away from each other, do not transform large, continuous tracts of undeveloped land like urban sprawl, agriculture, or forms of energy production that require surface mining. Thus, the impacts from wind energy on species, beyond collision fatalities, may be more likely to involve behavioral avoidance of tall structures, sound, or new human activity at the facility instead of local extirpation caused by habitat loss. In the US, the voluntary wind energy siting guidelines (Wildlife Service, 2016) consider ‘fragmentation sensitive species’ and the need to “Minimize the number and length of access roads; use existing roads when feasible”. Our results support this focus.

Other studies on energy and land transformation

Similar to Jones & Pejchar (2013) we found that the geographic context in which wind energy is developed changes the levels of impact. Using turbines in Colorado and Wyoming, Jones & Pejchar (2013) found fewer impacts on indicators of biodiversity and ecosystem services at individual turbines with more pre-construction development. We found similar patterns at facilities across the US and in a wide variety of vegetation types. We estimated wind facilities nearly doubled the length of roads (a 100% increase), while Jones & Pejchar (2013) estimated an increase of ∼250% (∼450 to ∼1,600 m of road, Fig. 4 of their paper). Both studies used a one km buffer. It is possible our study had a larger proportion of sites with high levels of pre-construction development and thus estimated lower levels of new roads. However, we digitized entire facilities and therefore included areas that had roads, but no new turbines, within the study extent. Jones & Pejchar (2013) centered each sampling location at a turbine, and thus always sampled an area that included a turbine and its newly built access road. This difference in sampling entire facilities versus individual turbines likely caused the differences in the predicted changes to roads from wind energy development in our studies.

The negative relationship between levels of pre-construction development and the amount of land change caused by the construction of wind facilities likely holds true for other forms of energy. Jones & Pejchar (2013) found this pattern for oil and gas wells. The strength of this relationship indicates it should be considered in broader scale or generalized analyses of the land transformation caused by energy (Fthenakis & Kim, 2009; McDonald et al., 2009; Trainor, McDonald & Fargione, 2016). For example, Trainor, McDonald & Fargione (2016) use a single value of ‘land use efficiency’ (km2/TWhr) to extrapolate the amount of land cover change caused by different energy types across the US under different development scenarios. Across the facilities we digitized, 39.0 ± 7.0% (mean ± SE) of the facility was located in already developed land. If a concern about energy development is the impacts to undeveloped lands, then perhaps we should not include parts of the facility that utilize existing developed land when calculating land use efficiency. Furthermore, extrapolations about future land use effects of energy will improve if they consider where that new energy infrastructure will be located relative to existing development.

Conclusions

Our results support the intuitive, but not well-tested, hypothesis that the geographic context in which energy is developed can alter the types and levels of impacts such development has on natural systems. In particular, we found levels of pre-construction development and new roads influenced the impacts of wind facilities on undeveloped land, and the expansion of road networks was further influenced by facility design. Our results suggest careful siting and planning of new facilities that optimize energy outputs while utilizing already disturbed locations and minimizing new roads may limit impacts. Based on these conclusions, we also expect that energy suitability analysis and comparisons of energy impacts across sources would benefit from the inclusion of geographic context. Even if one form of energy generates more land transformation per unit of energy produced, one may overlap endangered species and urban areas, while the other may not. These interactions between energy and the locations of other resources affect how we measure impacts and our ability to develop energy resources. Conflicts with other co-located resources and land uses may cause energy developments to violate legal standards, become too costly, or exceed the level of impact society is willing to accept, causing those energy resources to become ‘stranded.’ These effects could be anticipated and better accounted for by considering geographic context when quantifying energy development impacts.

Supplemental Information

Supplemental Information 1 Maps of 5 wind facilities before and after construction

Click here for additional data file.

Astute feedback from two anonymous reviewers and T. Conkling greatly improved the paper. Roger Compton (USGS, retired) performed the original digitizing of the post-construction facilities and we thank him for his work. Any use of trade, firm, or product names is for descriptive purposes only and does not imply endorsement by the US Government.

Additional Information and Declarations

Competing Interests

Author Contributions

Data Availability

The authors declare there are no competing interests.

Jay E. Diffendorfer and Monica A. Dorning conceived and designed the experiments, performed the experiments, analyzed the data, contributed reagents/materials/analysis tools, prepared figures and/or tables, authored or reviewed drafts of the paper, approved the final draft.

Jolene R. Keen and Louisa A. Kramer conceived and designed the experiments, performed the experiments, authored or reviewed drafts of the paper, approved the final draft, digitized wind facilities.

Robert V. Taylor performed the experiments, authored or reviewed drafts of the paper, approved the final draft, digitized wind facilities.

The following information was supplied regarding data availability:

Data is available at Sciencebase: Diffendorfer, JE, Dorning, MA, Keen, JR, Kramer, LA, Taylor, RV, 2019, Data release for Geographic context affects the landscape change and fragmentation caused by wind energy facilities: U.S. Geological Survey, https://doi.org/10.5066/F7639NZB.

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
