# Peer review of "Geographic context affects the landscape change and fragmentation caused by wind energy facilities"

_PeerJ, doi:10.7717/peerj.7129_

## Round 0.1 · original submission · Minor Revisions

The reviewers disagreed as to how well the paper described the method and the support for the conclusions. Although I lean more towards Reviewer 2, the authors should consider the points made by Reviewer 1 and how they can be made clearer. From a landscape ecology point of view, I found the digitisation process well enough described.

The most robust finding is that encapsulated in Figure 4. Reviewer 1 concluded that there were dubious relationships displayed in parts of Figures 2 and 3 - certainly on a visual basis, and is asking for more detail.

The part missing most from the explanation of method is how AICc was used and RVI assessed. Neither is defined in the text and should be. While it's clear that the authors tried to be parsimonious because they were using an information-weighted approach, this part needs to be better explained.

This is a political issue, and papers such as this will be looked at by different elements of industry with a range of ideological views. It may be used in contested planning and development proposals. For that reason your figures and explanations needs to be as defensible as you can make them.

The wide spread of variables across core area index and connectivity index does need to be considered and discussed. From an ecological or ecosystems services point of view, there are clearly many more factors that need to be considered on a site by site basis, and some of these could be mentioned to help put these results into a wider practical context.

The references inserted into the text need to be cleaned up (e.g., appropriate use of parentheses)

Reviewer 1 ·

Basic reporting

Details in attachement

Experimental design

Details in attachement

Validity of the findings

Details in attachement

Additional comments

Details in attachement

Annotated reviews are not available for download in order to protect the identity of reviewers who chose to remain anonymous.

Reviewer 2 ·

Basic reporting

This was a very well written paper on an interesting and very timely topic. The presentation was extremely clear and I have few comments.

Experimental design

The presentation was extremely clear and I have few comments. The assessment of variable importance and parameter estimates for the statistical models is solid and well done.

Validity of the findings

As renewable energy, and wind energy in particular, expands rapidly in response to both calls to reduce CO2 emissions as well as the levelized cost of electricity makes wind one of the cheapest forms of new electricity capacity – it will be critical to proactively consider siting and mitigation of wind energy impacts. Landscape-scale assessments like that described in this manuscript can be used to evaluate impacts needed to guide land-use planning and management decisions.

Additional comments

Comments to the authors of “Geographic context affects the landscape change and fragmentation caused by wind energy facilities” by Jay E Diffendorfer, Monica A Dorning, Louisa A Kramer, Jolene Keen, and Robert V Taylor.

This was a very well written paper on an interesting and very timely topic. The presentation was extremely clear and I have few comments. The assessment of variable importance and parameter estimates for the statistical models is solid and well done. This is a great example of the interface of landscape-scale assessments and field research in order to evaluate impacts of land-use decisions. I think it would be very interesting for a wide variety of readers and could help land managers see how science can be used to better guide land use decisions. As the authors state their results support the intuitive, but not well-tested, hypothesis that the geographic context in which energy is developed can alter the types and levels of impacts such development has on natural systems. As renewable energy, and wind energy in particular, expands rapidly in response to both calls to reduce CO2 emissions as well as the levelized cost of electricity makes wind one of the cheapest forms of new electricity capacity – it will be critical to proactively consider siting and mitigation of wind energy impacts. Landscape-scale assessments like that described in this manuscript can be used to evaluate impacts needed to guide land-use planning and management decisions.

As I stated I have few technical comments – most of the specific points that I think should be addressed relate to the interpretation and communication of the results.

My biggest concern of the manuscript is that I think it fails to include one of the most significant forms of impact associated with energy development: transmission. In many cases the infrastructure needed to support energy development has a larger impact than the development itself. For example, the US Department of Energy estimates that it will take approximately 5 million hectares to reach 20% electricity production. But it will also take an additional 18,000 kilometers of new transmission lines. The current study examines only the impacts of “onsite” transmission features when it is the “offsite” transmission that has the largest impact. It would be great to discuss how siting new wind installations near previously altered landscapes may further reduce the impacts from new development.

The authors ask the important question is it possible to construct facilities that minimize the levels of habitat loss and fragmentation? Their results do suggest siting facilities in locations with higher levels of existing development can reduce impacts. Several other studies have speculated as to the importance of siting renewable energy on lands already impacted from development in order to reduce impacts given projections of future wind energy development. It would however be good to link the current assessment to these other studies that have estimated this potential – Kiesecker et al. 2011 (US wide), Fargione et al. 2012 (Northern Great Plains), Baruch-Mordo et al. 2018 (Global).

I see a reference to Jones et al. but don’t see a Jones et al. in the literature cited.

I would like to see how the authors think the results of their analysis could be incorporated into application – more specific details here would be good.

External reviews were received for this submission. These reviews were used by the Editor when they made their decision, and can be downloaded below.

---

## Round 0.2 · Minor Revisions

Thanks for the re-submission with revisions. Although the reviewers' comments have been well addressed and the paper is clearer I'm asking for minor revisions to clean up the text because PeerJ does not copy edit the main text.

For all references, where the authors are mentioned in text, it should be author (date), not (author, date).

information theoretic modeling approach needs a hyphen, lines 115, 227

I would use proper mathematical operators in algorithms, not *

lower case table should be uc 368

lower case figure ditto 388 (there may be others)

superfluous comma the change before - line 200 (confuses a bit)

If you do all this which won't take long and will improve the quality of your final article, and the re-submission will be waved through

External reviews were received for this submission. These reviews were used by the Editor when they made their decision, and can be downloaded below.

---

## Round 0.3 · accepted · Accept

Thanks for doing those - all good to go!

External reviews were received for this submission. These reviews were used by the Editor when they made their decision, and can be downloaded below.